# High Sensitivity Photonic Crystal Fiber Refractive Index Sensor with Gold Coated Externally Based on Surface Plasmon Resonance

**DOI:** 10.3390/mi9120640

**Published:** 2018-12-03

**Authors:** Xudong Li, Shuguang Li, Xin Yan, Dongming Sun, Zheng Liu, Tonglei Cheng

**Affiliations:** 1State Key Laboratory of Synthetical Automation for Process Industries, College of Information Science and Engineering, Northeastern University, Shenyang 110819, China; lxd163youxiang@163.com (X.L.); lishuguang@ise.neu.edu.cn (S.L.); yanxin@ise.neu.edu.cn (X.Y.); dmsun@imr.ac.cn (D.S.); zhengliuyouxiang@163.com (Z.L.); 2Shenyang National Laboratory for Materials Science, Institute of Metal Research, Chinese Academy of Sciences, 72 Wenhua Road, Shenyang 110016, China

**Keywords:** Photonic crystal fiber, surface plasmon resonance, refractive index, sensor, finite element method

## Abstract

In this paper we propose a gold-plated photonic crystal fiber (PCF) refractive index sensor based on surface plasmon resonance (SPR), in which gold is coated on the external surface of PCF for easy fabrication and practical detection. The finite element method (FEM) is used for the performance analysis, and the numerical results show that the thickness of the gold film, the refractive index of the analyte, the radius of the air hole in the first layer, the second layer, and the central air hole can affect the sensing properties of the sensor. By optimizing the sensor structure, the maximum wavelength sensitivity can reach 11000 nm/RIU and the maximum amplitude sensitivity can reach 641 RIU^−1^. Due to its high sensitivity, the proposed sensor can be used for practical biological and chemical sensing.

## 1. Introduction

Surface plasmon resonance (SPR) is a new technology that has experienced rapid development in recent decades. Due to its unique capabilities such as high sensitivity and usefulness in unlabeled detection [1], SPR has been widely used in fields such as biological analysis [2,3], food quality detection [4], and chemical and gas analysis [5,6]. However, the traditional prism-based SPR sensors have many disadvantages, such as large volume and high cost [7], which naturally limit the practical application. Recently, photonic crystal fiber (PCF) based SPR sensor attracted much attention due to features such as small in size and low in loss. Compared with ordinary optical fibers, PCF is more flexible in design [8]. We can modify air holes geometries or alter the number of rings to manage the light propagation, and a good evanescent field can be obtained in PCF. Up to now, PCF has been widely applied in many fields such as lasers [9], demultiplexer [10], and demux [11].

In order to realize the PCF based SPR sensor, a metal film can be coated on the air hole inside the fiber [12,13]. For example, in 2014, Jitendra et al. put a layer of metal on six liquid analyte channels [14], and its maximum wavelength sensitivity reached 2000 nm/RIU. However, selective coating of metal films in micro pores is a complex process in fabrication. To overcome this shortcoming, a novel kind of PCF based SPR sensor [15,16] using nanowires is proposed. For example, Liu et al. put the gold nanowires into PCF in 2018. The sensor’s detection range is reported to be between 1.33 and 1.38, and the highest sensitivity is 4111 nm/RIU [17]. Nevertheless the process of filling nanowires and analytes into the required air holes is also a touch issue.

Compared with metal-coated-inside PCF SPR sensors and nanowire-based PCF SPR sensors, PCF SPR sensors coating metal and analyte outside of the fiber structure greatly simplifies the manufacture and measurement process. In this paper, a PCF based SPR sensor with gold coated on the external surface of PCF is proposed. By adjusting parameters such as the thickness of the metal film, the radius of the central air hole and the radius of the air hole in the cladding, the sensor structure is optimized and its performance is greatly enhanced. Its maximum wavelength sensitivity can reach 11,000 nm/RIU and the maximum amplitude sensitivity is 641 RIU^−1^ at the refractive index range of 1.390~1.395.

## 2. Structural and Theoretical Modeling

The cross section of the PCF sensor is shown in Figure 1. The radius of the central air hole is *r*_0_ = 0.2 μm. The air holes in the first ring have a radius of *r*_1_ = 0.4 μm, and they are arranged in a clockwise rotation of 60°. The distance between the center of the air holes and the center of the fiber is *d* = 2 μm. In the second ring, the air holes have a radius of *r*_2_ = 0.6 μm and they rotate clockwise at 30°. The distance between the first layer and the second is Λ = 1.2 μm, and two holes are missing in the opposite vertices of the second ring. The thickness of the gold layer is *t_g_* = 35 nm. Outside the gold layer is the analytical layer.

The background material of the proposed PCF sensor is silica. The dispersion relation can be obtained from the Sellmeier formula by Reference [18].
(1)$$n^{2}(\omega )=1+\sum_{j=1}^{3}\frac{B_{j}\lambda^{2}}{\lambda^{2}-\lambda_{j}^{2}}$$
where, *B*_1_ = 0.6961663, *B*_2_ = 0.4079426, *B*_3_ = 0.8974764, $\lambda_{1}$ = 0.0684043 μm, $\lambda_{2}$ = 0.1162414 μm, $\lambda_{3}$ = 9.896161 μm.

As gold has good oxidation resistance, it is chosen as the metal layer, whose thickness is represented by *t_g_*. The dielectric constant of gold can be expressed as Drude-Lorentz [19], which is expressed as follows:(2)$$\varepsilon_{m}=\varepsilon_{\infty }-\frac{\omega_{D}^{2}}{\omega (\omega +j\gamma_{D})}-\frac{∆\varepsilon \cdot \Omega_{L}^{2}}{(\omega^{2}-\Omega_{L}^{2})-j\Gamma_{L}^{\omega }}$$
in the formula, the high frequency dielectric constant is $\varepsilon_{\infty }$ = 5.967 and the weighted coefficient ∆ε = 1.09. ω is the guiding optical angular frequency, $\omega_{D}$ the plasmon frequency, $\gamma_{D}$ the damping frequency, $\Omega_{L}$ the oscillator strength of the Lorenz oscillator, and $\Gamma_{L}$ the frequency spectrum width of the Lorenz oscillator.

The performance of the proposed PCF sensor is analyzed by the finite element method (FEM) based on the COMSOL Multiphysics software, and we set a scattering boundary condition of the perfectly matched layer (PML) to absorb the radiation energy. The confinement loss is an important factor in measuring the performance of sensors. Which can be expressed as [20]:(3)α(dB/cm)=8.686×k0Im(neff)×104
where $k_{0}=2\pi /\lambda$ is the wave number and Im(neff) is the imaginary part of the refractive index.

## 3. Simulation Results and Analysis

The working mechanism of the PCF based SPR sensor is that the evanescent field produced by the light wave propagating in the core interacts with the plasma metal to produce surface plasmon wave (SPW). When light propagates in the core, its electromagnetic field leaks into the cladding area and excites free electrons on the surface of the plasma metal. SPR occurs when the leakage electromagnetic field matches the frequency of free electrons on the metal surface. Under resonance, the maximum energy is transferred from the core guidance mode to thesurface plasmon polaritons (SPP) mode. PCF consists of core layer and cladding. The refractive index of core and cladding can be adjusted to control the propagation of light and the electromagnetic field. We can optimize the sensor’s performance by adjusting them.

Figure 2a shows the distribution of the effective index of the proposed sensor. The blue curve indicates the refractive index of the core-guided mode and it decreases with the increase of wavelength. The red curve indicates the refractive index of the SPP mode that decreases faster than the former and intersects with it at the wavelength of 0.674 μm. The green curve indicates the confinement loss and it reaches the peak value at the resonance wavelength, which indicates that the phase matching condition is achieved. Figure 2b shows the optical field of the core-guided mode and it mainly distributes in the core. Figure 2c shows the optical field distribution of the SPP mode and it mainly distributes on the surface of gold film. Figure 2d shows the optical field of the core-guided mode and the SPP mode at the resonance wavelength. We can see that there is a strong coupling between their two modes, and the optical field distributes both on the surface of the gold film and in the core. This confirms the occurrence of the phase matching condition.

Figure 3 shows the effect of the thickness of the gold layer (*t_g_*) on the confinement loss of the core-guided mode. It can be seen from the figure that when *t_g_* increases from 30 to 50 nm, the resonance intensity decreases and the resonance wavelength red-shifts. This is because the thicker the *t_g_*, the higher the damping loss. As a result, the core-guided mode has less energy penetration into the gold film and the coupling effect becomes weaker.

Figure 4 shows the effect of the size of the central air hole radius (*r*_0_) on the confinement loss of the core-guided mode. It can be seen from the diagram that, when *r*_0_ changes from zero to 0.25 μm, the resonance intensity increases, and the resonance peak becomes sharper and the resonance wavelength shifts red. This is because the change of *r*_0_ affects the refractive index of the core-guided mode. Thus, the phase matching between SPP mode and the core-guided mode is affected, which in turn leads to a change in the resonance intensity, resonance peak, and the resonance wavelength. Therefore, we can change the size of *r*_0_ to optimize the sensor performance.

Figure 5 shows the influence of the size of the-first-layer air holes (*r*_1_) on the confinement loss of the core-guided mode. It can be seen that when *r*_1_ increases from 0.34 to 0.42 μm, the resonance intensity increases slowly and the resonance wavelength shifts red. This is because the change of *r*_1_ affects the refractive index of the cladding region, leading to the change of the phase matching condition. Figure 6 displays the influence of the size of the-second-layer air holes (*r*_2_) on the confinement loss of the core-guided mode. It can be seen that with the increase of *r*_2_ (from 0.57 to 0.69 μm), the resonance intensity decreases, the resonance peak becomes sharper, and the resonance wavelength shifts blue.

From the above analysis, we can see that the thinner the gold layer, the bigger the size of the central air hole and the first-layer air holes, the better the performance of the sensor. As to the size of the second -layer air holes, the smaller the better. However, if the thickness of the gold layer goes below 35 nm, or if the size of the central air hole goes beyond 0.2 μm, the confinement loss will be too large to detect. Taking this into consideration as well as the fabrication feasibility, the thickness of 35 nm is selected, and the size of the-first-layer air hole is set to be 0.4 μm and the size of the-second-layer air holes is 0.6 μm.

SPR is very sensitive to the change of the refractive index of the surrounding dielectric. When the refractive index of the analyte varies from 1.350 to 1.395, the loss spectra of the sensor are shown in Figure 7. It can be seen that the resonance intensity increases with the increase of refractive index of the analyte, the resonance peak becomes sharper, and the resonance wavelength shifts red. This is due to the fact that the real part of refractive index of SPP mode depends strongly on the refractive index of analyte. When the refractive index of the analyte changes, even slightly, real part of the refractive index of SPP mode changes. Wavelength sensitivity can be used to display the performance of the sensor, which can be expressed as follows:(4)$$S_{\lambda }(\mathrm{nm}/\mathrm{RIU})=∆\lambda_{peak}/∆n_{a}$$
where $∆\lambda_{peak}$ is the offset of the resonance peak under different refractive index and $∆n_{a}$ is the change of refractive index of different analytes.

When the refractive index of the analyte is 1.350, 1.355, 1.360, 1.365, 1.370, 1.375, 1.380, 1.385, 1.390, and 1.395, the resonant peak is 0.645, 0.659, 0.674, 0.691, 0.711, 0.734, 0.762, 0.796, 0.838, and 0.893 μm, respectively, which are shown in Figure 8. For the proposed sensor, when the refractive index of the analyte is between 1.350 and 1.395, an average spectral sensitivity of 5500 nm/RIU can be achieved. Particularly, when the refractive index range is between 1.390 and 1.395, its spectral sensitivity can reach as high as 11,000 nm/RIU.

The resolution of the sensor can be expressed as:(5)$$R(RIU)=∆n_{a}\times ∆\lambda_{\min}/∆\lambda_{peak}$$
where $∆\lambda_{\min}$ is the smallest spectral resolution. Assuming $∆\lambda_{\min}$ is 0.1 nm, the maximum resolution of the sensor is calculated to be 9.09 × 10^−5^ RIU.

Table 1 presents the performance comparison of the previously reported sensors and our proposed sensor. It can be seen from the table that the of our sensor is more sensitive, which indicates that it is more applicable in practical biological and chemical sensing.

The amplitude interrogation method is often applied in practice, which is due to the fact that it can be measured at specific wavelengths and independently of spectrum operation. Additionally, the amplitude sensitivity of the sensor is shown in Figure 9, which can be defined as follows:(6)$$S_{A}(RIU^{-1})=-\frac{1}{\alpha (\lambda ,n_{a})}\frac{\partial \alpha (\lambda ,n_{a})}{\partial n_{a}}$$
where $\alpha (\lambda ,n_{a})$ is the confinement loss at the analyte refractive index of *n_a_*, $\partial \alpha (\lambda ,n_{a})$ is the difference of confinement loss due to two adjacent refractive indexes of two analytes. When the refractive index of the analyte is 1.390, the maximum amplitude sensitivity of the sensor can reach 641 RIU^−1^ at the wavelength of 0.893 μm.

.

## 4. Conclusions

A PCF based SPR sensor was presented in the paper, whose performance was analyzed by FEM. The thickness of the gold film, the refractive index of the analyte, the radius of the first, the second layer air holes, and the center air hole are analyzed and their influence on the sensing property are presented in detail. After the optimization of the fiber structure, we have adopted the following design parameters: *r*_0_ = 0.2 μm, *r*_1_ = 0.4 μm, *r*_2_ = 0.6 μm, Λ = 1.2 μm, *d* = 2 μm, and *t_g_* = 35 nm. Additionally, the maximum wavelength sensitivity of the sensor can reach as high as 11,000 nm/RIU and the amplitude sensitivity 641 RIU^−1^ at the refractive index range of 1.390~1.395. The proposed sensor is simple, reliable, easy to fabricate, and highly applicable, and it can be used for practical biological and chemical sensing.

## Figures and Tables

**Figure 1 micromachines-09-00640-f001:**
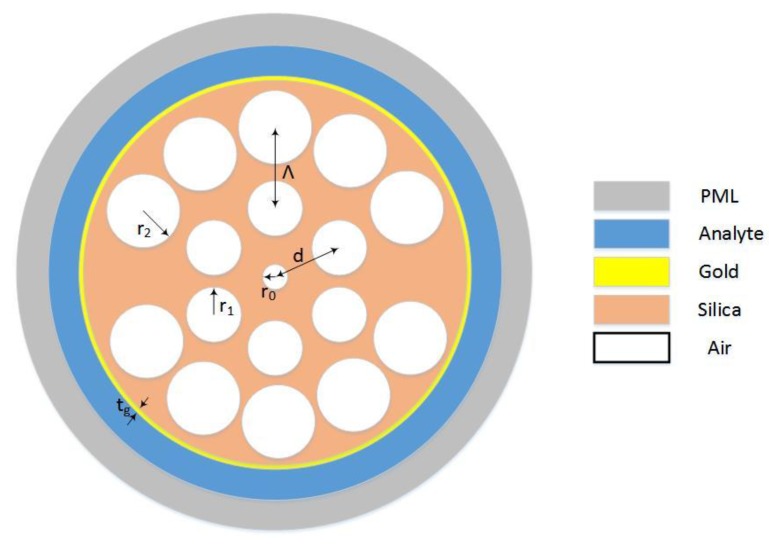
Cross-section of the proposed photonic crystal fiber (PCF) sensor.

**Figure 2 micromachines-09-00640-f002:**
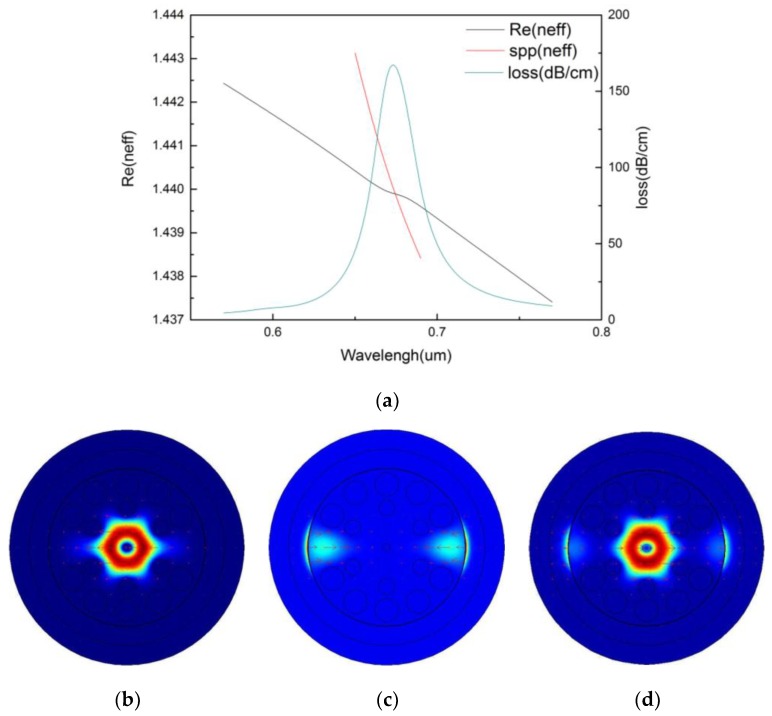
(**a**) Dispersion relation between surface plasmon polaritons (SPP) mode and core-guided mode; optical field distribution of (**b**) the core-guided mode, (**c**) the SPP mode, and (**d**) Optical field distribution in phase matching. $\left(n_{a}=1.36,r_{0}=0.2\;\mu m,r_{1}=0.4\;\mu m,r_{2}=0.6\;\mu m,\Lambda =1.2\;\mu m,d=2\;\mu m,t_{g}=35\;\mathrm{nm}\right)$.

**Figure 3 micromachines-09-00640-f003:**
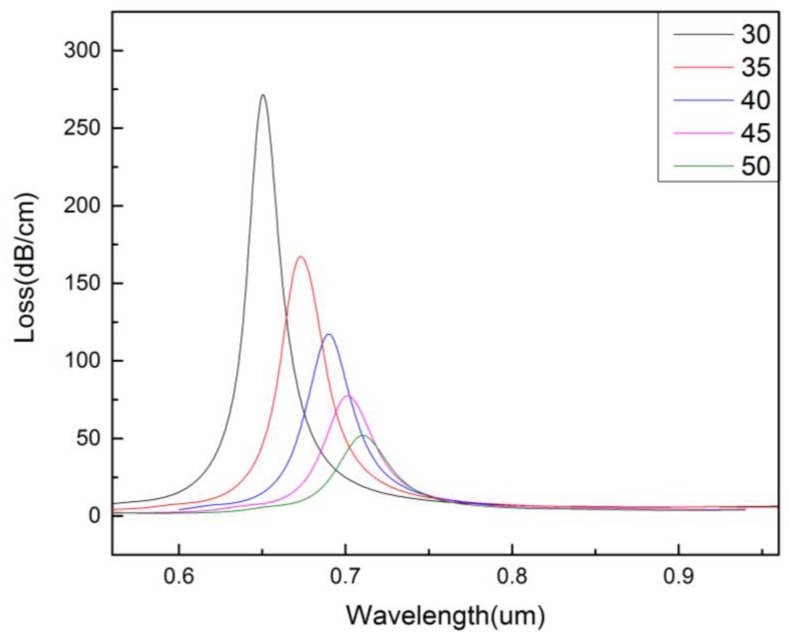
The confinement loss of the core-guided mode for different thickness of the gold layer. $\left(n_{a}=1.36,r_{0}=0.2\;\mu m,r_{1}=0.4\;\mu m,r_{2}=0.6\;\mu m,\Lambda =1.2\;\mu m,d=2\;\mu m\right)$.

**Figure 4 micromachines-09-00640-f004:**
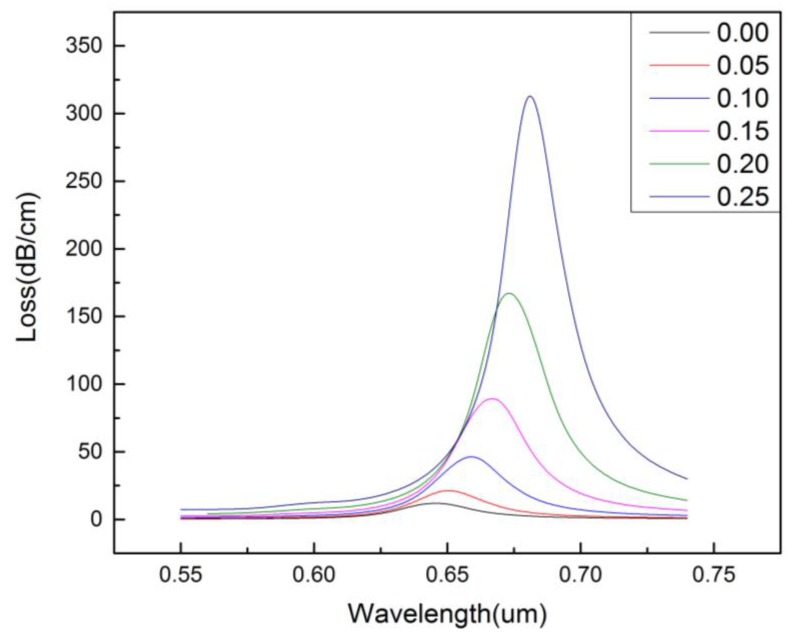
The confinement loss of the core-guided mode for different size of the central air hole. $\left(n_{a}=1.36,r_{1}=0.4\;\mu m,r_{2}=0.6\;\mu m,\Lambda =1.2\;\mu m,d=2\;\mu m,t_{g}=35\;\mathrm{nm}\right)$.

**Figure 5 micromachines-09-00640-f005:**
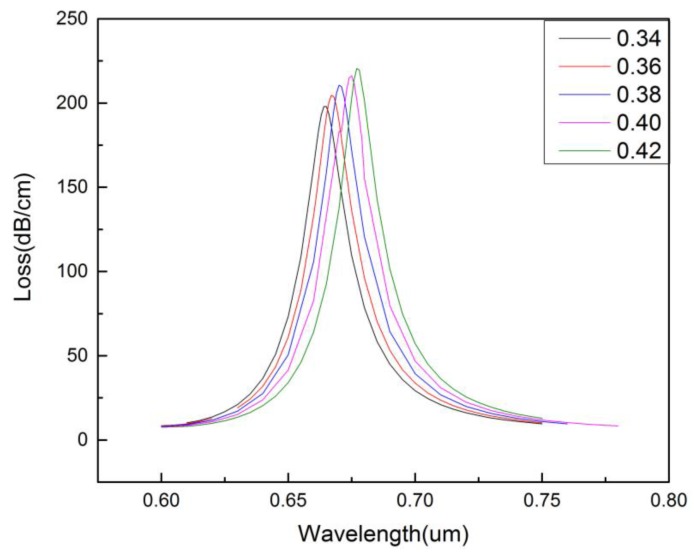
The confinement loss of the core-guided mode for different size of the-first-layer air holes. $\left(n_{a}=1.36,r_{0}=0.2\;\mu m,r_{2}=0.6\;\mu m,\Lambda =1.2\;\mu m,d=2\;\mu m,t_{g}=35\;\mathrm{nm}\right)$.

**Figure 6 micromachines-09-00640-f006:**
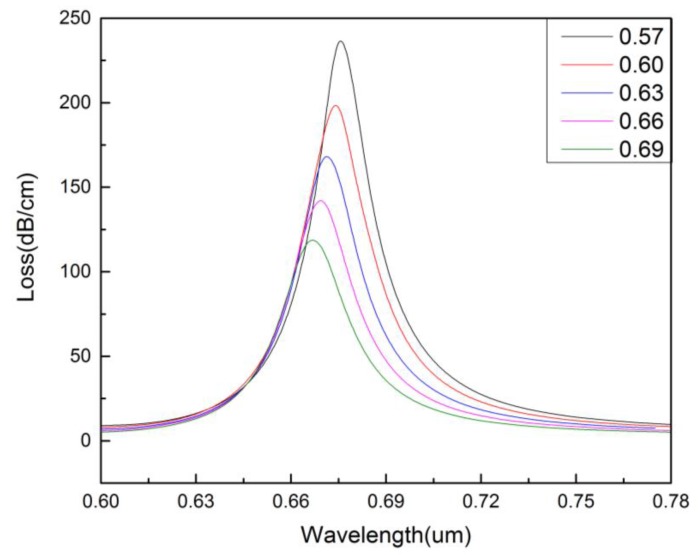
The confinement loss of the core-guided mode for different size of the-second-layer air holes. $\left(n_{a}=1.36,r_{0}=0.2\;\mu m,r_{1}=0.4\;\mu m,\Lambda =1.2\;\mu m,d=2\;\mu m,t_{g}=35\;\mathrm{nm}\right)$.

**Figure 7 micromachines-09-00640-f007:**
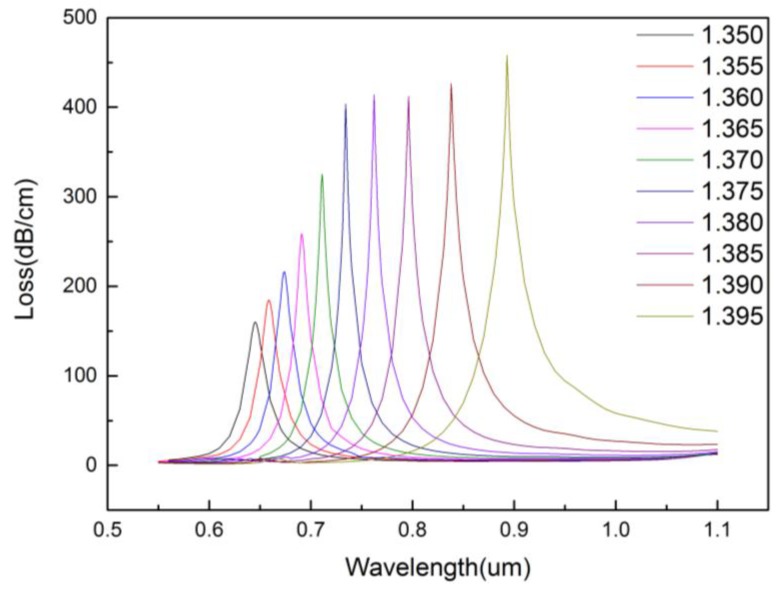
Loss spectral when increasing analyte RI from 1.350 to 1.395. $\left(r_{0}=0.2\;\mu m,r_{1}=0.4\;\mu m,r_{2}=0.6\;\mu m,\Lambda =1.2\;\mu m,d=2\;\mu m,t_{g}=35\;\mathrm{nm}\right)$.

**Figure 8 micromachines-09-00640-f008:**
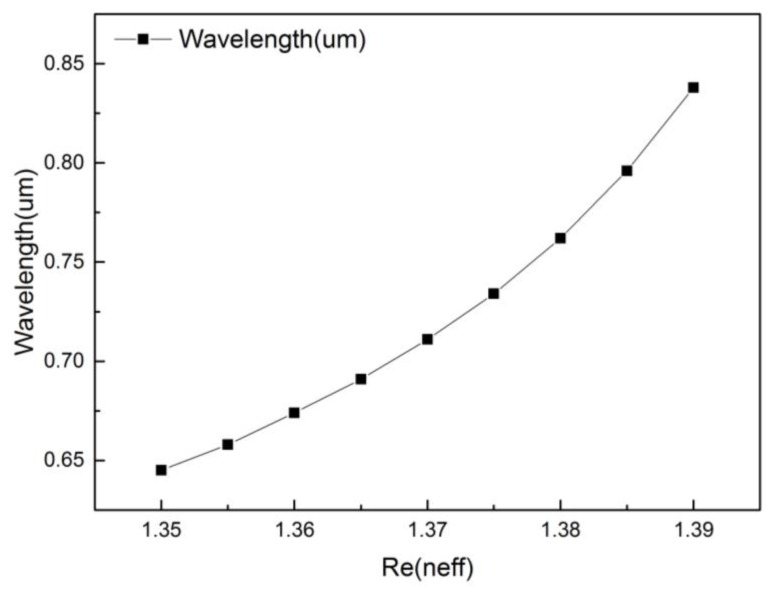
The resonance peak when increasing analyte Re from 1.350 to 1.395. $\left(r_{0}=0.2\;\mu m,r_{1}=0.4\;\mu m,r_{2}=0.6\;\mu m,\Lambda =1.2\;\mu m,d=2\;\mu m,t_{g}=35\;\mathrm{nm}\right)$.

**Figure 9 micromachines-09-00640-f009:**
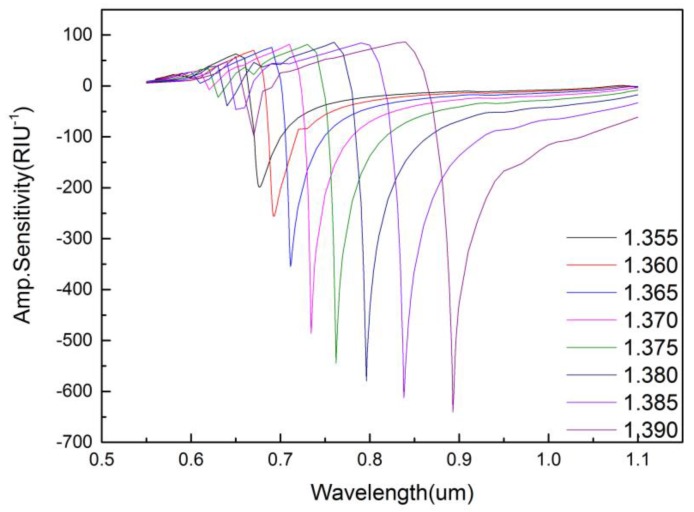
Amplitude sensitivity when increasing analyte RI from 1.350 to 1.395. $\left(r_{0}=0.2\;\mu m,r_{1}=0.4\;\mu m,r_{2}=0.6\;\mu m,\Lambda =1.2\;\mu m,d=2\;\mu m,t_{g}=35\;\mathrm{nm}\right)$.

**Table 1 micromachines-09-00640-t001:** Performance comparison of the previously reported PCF-SPR sensors and the proposed sensor.

The Structure of PCF	Detection RI Range	Maximum Sensitivity
Double core structure [2]	1.35–1.36	2200 nm/RIU
Graphene-Based structure [21]	1.345–1.350	3400 nm/RIU
Hollow-core silver coated structure [22]	1.36–1.37	4200 nm/RIU
Double core structure [23]	1.33–1.34	4000 nm/RIU
Double core structure [24]	1.36–1.37	9000 nm/RIU
Structure with Gold Coated Externally (our work)	1.390–1.395	11,000 nm/RIU

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
