# Peer review of "High Sensitivity Photonic Crystal Fiber Refractive Index Sensor with Gold Coated Externally Based on Surface Plasmon Resonance"

_micromachines, 2018, doi:10.3390/mi9120640_

Round 1

Reviewer 1 Report

This manuscript theoretically investigate the sensing properties of externally gold-coated PCF by varying the geometric parameters.
1. The high sensing sensitivity is due to the coupling between the core-guided mode and SPR mode. This concept is not new. Similar paper have been published in 

A.A. Rifat, G.A. Mahdiraji, Y.M. Sua, Y.G. Shee, R. Ahmed, D.M. Chow, et al.Surface plasmon resonance photonic crystal fiber biosensor: a practical sensing approach

Photonics Technol. Lett. IEEE, 27 (2015), pp. 1628-1631

2. Higher sensing sensitivity as 46000nm/RIU can be obtained by using D-shape PCF as demonstrated in

A. A. Rifat, R. Ahmed, G. A. Mahdiraji, F. R. M. Adikan, "Highly sensitive D-shaped photonic crystal fiber-based plasmonic biosensor in visible to near-IR", IEEE Sensors J., vol. 17, no. 9, pp. 2776-2783, May 2017.

3.In Fig. 7, the reason for the linewidth variation should be discussed.

Author Response

Authors' Responses to Editor and Reviewer

Dear Editor and Reviewers,

   We appreciate the editor and the refereesinsightful and valuable comments. We have carefully revised the manuscript according to the suggestions and comments. Detailed point to point responses and changes are summarized as below. The revisions in the manuscript have been marked in red text.

Response to Reviewer 1

Reviewer #1 (Reviewer Comments Required):

This manuscript theoretically investigate the sensing properties of externally gold-coated PCF by varying the geometric parameters.

1. The high sensing sensitivity is due to the coupling between the core-guided mode and SPR mode. This concept is not new. Similar paper have been published in A.A. Rifat, G.A. Mahdiraji, Y.M. Sua, Y.G. Shee, R. Ahmed, D.M. Chow, et al. Surface plasmon resonance photonic crystal fiber biosensor: a practical sensing approach Photonics Technol. Lett. IEEE, 27 (2015), pp. 1628-1631

Respond: The performance of the sensor designed in this manuscript is much better than that published on the Photonics Technol. Lett. Using wavelength and amplitude interrogation methods, the proposed sensor in this manuscript can provide maximum sensitivity of 11000 nm/RIU and 641 RIU-1 , respectively, while the sensor published on the Photonics Technol. Lett can only provide that of 4000 nm/RIU and 320 RIU-1 . The comparison of these two sensors’performance is provided in Table 1 in the manuscript. 

2. Higher sensing sensitivity as 46000nm/RIU can be obtained by using D-shape PCF as demonstrated in A. A. Rifat, R. Ahmed, G. A. Mahdiraji, F. R. M. Adikan, "Highly sensitive D-shaped photonic crystal fiber-based plasmonic biosensor in visible to near-IR", IEEE Sensors J., vol. 17, no. 9, pp. 2776-2783, May 2017.

Respond: Although the sensitivity of the D-shape PCF proposed in IEEE Sensors is high, it required accurate polishing of the sensor surface to specifically eliminate a prearranged structure of the PCF. And it also needs to coat Au and TiO2 thin layers on the top of the D shaped PCF plane. This process is very tough. For our proposed sensor, this problem does not exist. The structure of our proposed sensor is simpler and it only requries to coat Au on the external surface of PCF. It is simple, reliable, easy operation and can realize real-time detectionbe.

3.In Fig. 7, the reason for the linewidth variation should be discussed.

Respond: Thank you for your advice. This is due to that the real part of refractive index of SPP mode depends strongly on the refractive index of the analyte. When the refractive index of the analyte changes, even slightly, the real part of the refractive index of SPP mode will change, leading to the change of the phase matching condition, which in turn leads to the linewidth changes. We have added the explanation in the third part of the manuscript.

Reviewer 2 Report

This paper presents simulation result for a fiberoptic SPR based refractive index sensor. The optimized structure demonstrated very high RI sensitivity. The simulations are sound and the results are interesting and I recommend the manuscript for publication. However, the authors should address the following points in their revision:

Line 46-47 and line 160-161: The authors claim sensitivity 11000nm/RI in the RI range 1.35-1.395. In fact the results presented on line 130-132 shows that this high sensitivity is achieved only in the range 1.39-1.395. In the lower end of the RI scale the sensitivity is closer to 3000nm/RI. 

The same goes for the numbers given in Table 1. Here it is important to note that ref [20] give the sensitivity in the RI range 1.34 to 1.37.  For the current study the sensitivity at RI 1.37 is about 4000nm/RI. We note that fig 8 in ref [20] shows a sensitivity of 4400nm/RI rather than 9000nm/RI. Perhaps this should be commented. 

The authors should add some argument for the chosen fiber design.

They should provide a reference for eqn (3) and give a short explanation

The results in figures 3, 4, 5, 6 are used to arrive at the final optimmized design. However, none of the figures actually shown any optimum. E.g. in Fig 3 the loss increase monotonically from 50 to 35 nm. What happens if you reduce the thickness below 35 nm? Similarly, in Fig 4  the loss increase monotonically from 0 to 0.2 um. What happens beyond 0.2um? And in Fig 5 the loss increase monotonically from 0.34 to 0.4 um. What happen above 0.4 um? In FIg 6 the loss increase monotonically from 0.69 to 0.6 um. What happens below 0.6 um?

On line 130 - 132: Why not present these results in a figure? This gives a much better insight into the resonant wavelength shift with RI. 

Figure 2 c) and d) are both labelled b).

On line 95: "As a result The core-guided mode..” should be lower case t.

Ref [20] is incomplete.

Author Response

Response to Reviewer 2
Reviewer #2 (Reviewer Comments Required):

This paper presents simulation result for a fiberoptic SPR based refractive index sensor. The optimized structure demonstrated very high RI sensitivity. The simulations are sound and the results are interesting and I recommend the manuscript for publication. However, the authors should address the following points in their revision:

Line 46-47 and line 160-161: The authors claim sensitivity 11000nm/RI in the RI range 1.35-1.395. In fact the results presented on line 130-132 shows that this high sensitivity is achieved only in the range 1.39-1.395. In the lower end of the RI scale the sensitivity is closer to 3000nm/RI.

The same goes for the numbers given in Table 1. Here it is important to note that ref [20] give the sensitivity in the RI range 1.34 to 1.37.  For the current study the sensitivity at RI 1.37 is about 4000nm/RI. We note that fig 8 in ref [20] shows a sensitivity of 4400nm/RI rather than 9000nm/RI. Perhaps this should be commented.

Respond: Thank you for your comment. We have modified the refractive index range of our proposed sensor to 1.390-1.395, and the refractive index range of ref [20] at the highest sensitivity is modified to 1.36-1.37. Thank you very much.

The authors should add some argument for the chosen fiber design.

Respond: Thank you for your advice. We have added the following in the first part of the manuscript.

Compared with ordinary optical fibers, PCF is more flexible in design [8]. We can modify air hole geometries and alter the number of rings in PCF to manage the light propagation. Apart from this, a good evanescent field can be obtained in PCF.

They should provide a reference for eqn (3) and give a short explanation

Respond: Thank you for your advice. The reference has been added and we have given a short explanation in the manuscript.

The results in figures 3, 4, 5, 6 are used to arrive at the final optimmized design. However, none of the figures actually shown any optimum. E.g. in Fig 3 the loss increase monotonically from 50 to 35 nm. What happens if you reduce the thickness below 35 nm? Similarly, in Fig 4 the loss increase monotonically from 0 to 0.2 um. What happens beyond 0.2um? And in Fig 5 the loss increase monotonically from 0.34 to 0.4 um. What happen above 0.4 um? In FIg 6 the loss increase monotonically from 0.69 to 0.6 um. What happens below 0.6 um?

Respond: Thank you for this question. We have added the cases when the thickness is 30 nm, the size of the central air hole radius 0.25 um, the size of the-first-layer air hole 0.42 um and the size of the-second-layer air hole 0.57 um in the manuscript. We can see that these simulations are in line with our finding.

However, if the thickness goes below 35nm, or if the size of the central air hole goes beyond 0.2um, the confinement loss will be too large to detect. Taking this into consideration as well as the fabrication feasibility, the thickness of 35nm is selected, and the size of the-first-layer air hole is set to be 0.4um and the size of the-second-layer air holes 0.6um. The relevant expression has been revised in the manuscript.

On line 130 - 132: Why not present these results in a figure? This gives a much better insight into the resonant wavelength shift with RI.

Respond: Thank you for your advice. We have added Figure 8 to present these results.

Figure 2 c) and d) are both labelled b).

On line 95: "As a result The core-guided mode.. should be lower case t.

Ref [20] is incomplete.

Respond: I am sorry for this negligence. It has been modified in the manuscript.

Reviewer 3 Report

In this paper, the authors introduced a gold-plated photonic crystal fiber (PCF) refractive index

 sensor based on surface plasmon resonance (SPR), in which gold is coated on the external surface of PCF for easy fabrication and practical detection. It shows some interesting results. The work is not acceptable in its present form, however, before the publication, there are few questions and suggestions.  The final decision depends on the authors' response.

1.The information for the PCF structure is not enough. The authors should give much more information about this. For example, they can add the x-z refractive index profile, So the readers can get its reproducibility. 

2.  The authors should give much more information about the novelty of this paper, especially the effect of using PCF why not used a classical fiber?

3. Did the authors simulate this device using an BPM solver or Band-solve solver in order to find the optimal geometrical values of the PCF and to see the photonic energy bands. Also, authors need to add in the results section the tolerances results for the key parameters (pitch, air-hole) which give much more data for the fabrication analysis,

4. More references need to be included in the introduction part to understand the applications and the light coupling in PCF (TIR and photonic bandgap)

"Prospects      for diode pumped alkali atom based hollow core photonic crystal fiber      lasers",Optics Letters, 39(16), 2014 (4655-4658).

b.       “An Eight-Channel C-Band Demux Based on Multicore Photonic Crystal Fiber”, Nanomaterials, vol. 8, issue. 845, OCT 2018.

c.       " A visible light RGB wavelength demultiplexer based on silicon-nitride multicore PCF, " Optica&Laser Technology, vol. 11, pp. 411-416, 2019.

5. The authors claim that this type of PCF It’s easy to fabricated (The proposed sensor is simple,

 reliable, easy to fabricate and highly applicable, and it can be used for practical biological and

 chemical sensing). However, if it easy why not to fabricated it?  this type of fiber its very complicated to fabricated because it very hard to processing gold and other materials and with a different air-hole size. Thus, this conclusion must be modified and authors needs to explain deeply how this fiber can be fabricated.

6.  Much more discussion about the results should be given in this paper, especially the author needs to provide enough physicals mechanism analysis about the results.

Author Response

Response to Reviewer 3

Comments and Suggestions for Authors

In this paper, the authors introduced a gold-plated photonic crystal fiber (PCF) refractive index sensor based on surface plasmon resonance (SPR), in which gold is coated on the external surface of PCF for easy fabrication and practical detection. It shows some interesting results. The work is not acceptable in its present form, however, before the publication, there are few questions and suggestions.  The final decision depends on the authors' response.

1.The information for the PCF structure is not enough. The authors should give much more information about this. For example, they can add the x-z refractive index profile, So the readers can get its reproducibility. 

Respond: Thank you for your advice. In the manuscript the presentation of our proposed PCF sensor is two-dimensional. We present the cross section in Figure 1, and we have adopted the following design parameters : r0 = 0.2um, r1 = 0.4um, r2 = 0.6 um, = 1.2um, d= 2um and tg =35nm. We have added the detailed information in the second part of the manuscript.

2. The authors should give much more information about the novelty of this paper, especially the effect of using PCF why not used a classical fiber?

Respond: Compared with ordinary optical fibers, PCF is more flexible in design. We can modify air hole geometries and alter the number of rings in PCF to manage the light propagation. We have added the relevant information in the first part of the manuscript.

3. Did the authors simulate this device using an BPM solver or Band-solve solver in order to find the optimal geometrical values of the PCF and to see the photonic energy bands. Also, authors need to add in the results section the tolerances results for the key parameters (pitch, air-hole) which give much more data for the fabrication analysis,

Respond: We analyze the performance of the proposed PCF sensor by the finite element method (FEM) based on the COMSOL Multiphysics softwar, and we have added the key parameters in the result part of the manuscript.

We have adopted the following design parameters: r0 = 0.2μm, r1 = 0.4μm, r2 = 0.6μm, = 1.2μm, d = 2μm and tg =35 nm.

4. More references need to be included in the introduction part to understand the applications and the light coupling in PCF (TIR and photonic bandgap)

"Prospects for diode pumped alkali atom based hollow core photonic crystal fiber      lasers",Optics Letters, 39(16), 2014 (4655-4658).

b.“An Eight-Channel C-Band Demux Based on Multicore Photonic Crystal Fiber”, Nanomaterials, vol. 8, issue. 845, OCT 2018.

c." A visible light RGB wavelength demultiplexer based on silicon-nitride multicore PCF, " Optica&Laser Technology, vol. 11, pp. 411-416, 2019.

Respond: Thank you for your advice. And we have added the references in the introduction part.

5. The authors claim that this type of PCF It’s easy to fabricated (The proposed sensor is simple, reliable, easy to fabricate and highly applicable, and it can be used for practical biological and chemical sensing). However, if it easy why not to fabricated it?  this type of fiber its very complicated to fabricated because it very hard to processing gold and other materials and with a different air-hole size. Thus, this conclusion must be modified and authors needs to explain deeply how this fiber can be fabricated.

Respond: Thank you for your comment. Compared with metal-coated-inside PCF SPR sensors and nanowire-based PCF SPR sensors, our proposed PCF sensor does simplify some operation steps. For example, gold is coated on the external surface of PCF instead of selective coating of metal films in micro pores. Chemical vapor deposition (CVD) method is reported as a possible coating method in a circular surface, and different air pressure can be added to the corresponding air holes in order to fabricate PCF with a different air-hole size. We have done a lot of experiments to fabricate PCF, and made a variety of photonic crystal fibers. The PCF fabrication has been preliminarily successful, but our technology of gold coating is not mature and there are still difficulties to overcome.

6. Much more discussion about the results should be given in this paper, especially the author needs to provide enough physicals mechanism analysis about the results.

Respond: Thank you for your advice. We have added physicals mechanism analysis about the results in the third part.

When light propagates in the core, its electromagnetic field leaks into the cladding area and excites free electrons on the surface of the plasma metal. SPR occurs when the leakage electromagnetic field matches the frequency of free electrons on the metal surface. Under resonance, the maximum energy is transferred from the core guidance mode to the SPP mode. PCF consists of core layer and cladding, whose refractive index can be adjusted to control the propagation of light and the electromagnetic field. We can optimize the sensor's performance by adjusting them.

Round 2

Reviewer 3 Report

Authors have been modified the paper and now it can send for publication.